# Processing and Tribological Properties of PEO Coatings on AlZn5.5MgCu Aluminium Alloy with Incorporated Al-Cu-Fe Quasicrystals

**Elena V. Torskaya** [1,*], **Alexei V. Morozov** [1,*] [ID], **Vladimir N. Malyshev** [2] [ID] **and Olga O. Shcherbakova** [1]

1 Ishlinsky Institute for Problems in Mechanics RAS, 119526 Moscow, Russia
2 Department—Tribology and Repair Technologies of Oil and Gas Equipment, Faculty of Engineering Mechanics, National University of Oil and Gas "Gubkin University", 119991 Moscow, Russia
* Correspondence: torskaya@mail.ru (E.V.T.); morozovalexei@mail.ru (A.V.M.)

**Abstract:** The modification of ceramic PEO coatings developed for friction units is in trend of modern research aimed at the improvement of friction and wear. This study presents ceramic coatings formed on AlZn5.5MgCu aluminum alloy by microarc oxidation (PEO) in a one-stage technological process with inclusions of Al-Cu-Fe quasi-crystals. The technology of rubbing quasi-crystals into the surface of the coating during final polishing was also used. Friction and wear tests, followed by analysis of the surface and cross-sections by SEM microscopy, showed that quasicrystals affect the coating structure and frictional characteristics. The addition of a small amount of modifier to the electrolyte (0.5 percent), as well as rubbing-in technology, can reduce the coefficient of friction (up to 16 percent) and wear rate (up to 43 times) compared to pure ceramic coatings. More modifier provides a negative result.

**Keywords:** ceramic coatings; Al-Cu-Fe quasicrystals; friction; wear

## 1. Introduction

Modification with quasicrystals is a promising method for improving the performance properties of various materials, in particular to reduce friction and increase wear resistance [1,2]. Quasi-crystals are effective in coatings obtained by various methods: high-velocity gas spraying [3,4], electron beam physical vapor deposition (PVD) [5,6], plasma spraying [7–10], laser melting [9,11], etc. The properties of the coatings with inclusions of quasicrystals are significantly improved, in particular the friction coefficient is noticeably reduced [12].

As is known, quasicrystals are complex intermetallic compounds with long-range quasiperiodic order [13]. The physical properties of quasicrystals differ from conventional crystalline and amorphous phases. High hardness, good wear resistance and low coefficient of friction offer prospects for use as a material for various coatings, as well as for the creation of composite materials by powder metallurgy methods [14,15]. Icosahedral quasicrystals of the Al-Cu-Fe system are aperiodic in three spatial directions and have unique properties that significantly improve the properties of composites [16].

To improve the wear resistance and mechanical properties of the surface of aluminum alloys, the method of microarc (plasma-electrolytic) oxidation (MAO/PEO) [17–22] is successfully used. The method, in contrast to the known process of hard anodization [23], is characterized by the fact that the process of coating formation occurs with the participation of microarc/arc discharges randomly moving over the treated surface. The temperature in the zone of the microarc discharge is about 4000–12,000 K [24], which makes it possible to oxidize the surface layer to oxides and spinels of various stoichiometric compositions, including higher aluminum oxide in the alpha modification (corundum). It provides high

physical, mechanical and strength properties of PEO coatings [25], wear resistance [26], heat resistance [27,28] and corrosion resistance [29], high electrical insulating properties, etc.

One of the advantages of the PEO technology is the possibility of forming coatings from suspension electrolytes [30,31]. The electrolytes contain, in addition to the components necessary for reproducing the process, a solid phase in the form of finely dispersed or nanosized particles of various nature, for example, oxides, carbides, borides, nitrides of metals [32]. The influence of the solid phase is very significantly manifested both on the properties of the coating being formed, and effectively increases the productivity of the formation process.

The aim of this study is to analyze the influence of the solid phase in the form of finely dispersed particles of quasicrystals of the AL-Cu-Fe system on the properties of PEO coatings. The idea of modifying PEO coatings with quasicrystals is new. It is based on the ability of quasicrystals to reduce friction, which is important for improving the antifriction properties of coatings intended for friction units. An integrated approach that combines the methods of experimental tribology and microscopy makes it possible to analyze the effect of quasicrystals on friction and wear of the PEO coatings.

## 2. Materials and Methods

### 2.1. Coatings Deposition

The coatings were obtained by the method of microarc oxidation [33]. A high-strength aluminum alloy of the AlZn5.5MgCu system (commercially available V95 alloy, GOST 21488-97, Russia) was used in the process. For the coating modification, the base electrolyte was doped by finely dispersed particles of quasicrystals of the Al (45.42% wt.) -Cu (32.9% wt.) -Fe (21.68% wt.) system [1,13] produced by Nanocom LLC (Skolkovo, Russia) [34]. The quasicrystal system had the following main technical properties: the dispersion of the base powder 1–60 μm; temperature stability in a non-oxidized environment 850–900 °C (600 °C intense oxidation in air); density 4000 kg/m$^3$; hardness 800–1000 HV; thermal conductivity 2 W·m$^{-1}$·K$^{-1}$. The particles of quasicrystals have an irregular polyhedral fragmented shape (Figure 1a). PEO disc samples with dimensions of Ø24 × 8 mm were pre-treated on a cast-iron plate with flatness control, cleaned of dirt, washed and dried in air.

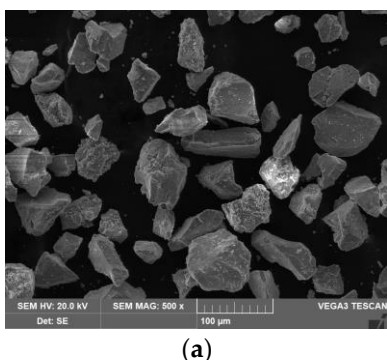
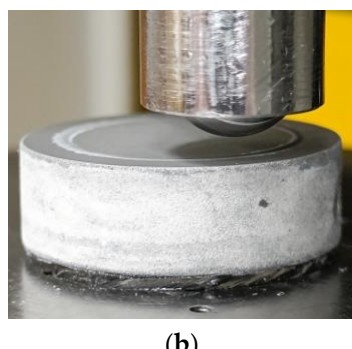

(**a**)  (**b**)

**Figure 1.** SEM image (**a**) of Al-Cu-Fe particles and a photo (**b**) of the disc with PEO coating after friction test and WC ball in the holder.

Slightly alkaline electrolyte (2 g/L KOH + 9 g/L Na$_2$SiO$_3$9H$_2$O + 6 g/L Na$_6$P$_6$O$_{18}$) was supplemented with various amounts of quasicrystal particles (5, 10, and 15 g/L). All used chemicals were with "Pure" qualification and produced in Russia in accordance to GOST 9285-78 (KOH), GOST 13078-81 (Na$_2$SiO$_3$9H$_2$O) and GOST 20291-80 (Na$_6$P$_6$O$_{18}$), respectively. At least 4 identical samples were formed for friction and wear tests.

The process of coating formation occurred in an electrolytic bath made of stainless steel with a cooling chamber with a capacity of 2 L, which served as a counter electrode. The process lasted for 60–70 min at a current density of 10 A/dm$^2$ using a capacitor-type

power supply [18,35]. It provides the anode–cathode mode of formation with the ratio of the cathode and anode currents Ic/Ia = 1.0–1.3.

The temperature of the microarc discharges at the moments of their burning (fractions of a second), as the authors point out [24,36], really reaches 4000–12,000 K, distracting for the quasicrystals. However, on the surface of the aluminum alloy, it is not so high due to the electrolyte cooling. As a result, it is possible to say that the quasicrystals on the surface of the sample should be in a molten form and be included in the composition of the oxides constituent components of the quasicrystals (Al-Cu-Fe) only when the particles of the quasicrystal fall into the region of operation of the microarc discharge. The main volume of quasicrystals particles, under the influence of both electrophoretic forces and electrolyte bubbling, accumulates mainly in the pores of the coating, or is embedded in the lattice of crystalline of the formed coating, drawn down into the discharge burning zone.

After the formation of coatings, the samples were washed in running water, dried with compressed air, and polished on a cast-iron plate using diamond paste and kerosene to a surface roughness of less than Ra = 2.5 μm. The polishing process was carried out manually. As an alternative to using quasi-crystals in the electrolyte, the quasicrystals were rubbed into the surface of the coatings formed without a modifier. The rubbing method is similar to the polishing process, but after the main step of polishing, a quasicrystal powder was used instead of diamond paste with kerosene. The average rubbing time was 3 min.

After that, the samples were washed again with detergents under running water and dried, and the thickness of the coatings was measured using MiniTest 600 FN2 (Elktro-Physik, Cologne, Germany).

Coating hardness was estimated using Vickers micro-hardness tester NanoScan 4D (Tisnum, Troitsk, Russia) under a load of 0.100 kg at 10 s. Ten measurements were made for each sample.

Thus, five variants of samples with PEO coatings were obtained (see Table 1): coatings without modifier (1), coatings with three different concentrations of quasicrystals in the electrolyte (0.5, 1.0, and 1.5 percent for Samples 2–4, respectively), coatings with quasicrystals rubbed into the surface applied during finishing (5).

**Table 1.** The sample code and main properties of PEO coatings.

| Sample Codes | Quasicrystal Modifier, $g \cdot L^{-1}$ | Roughness, mkm | | Average Thickness, μm | Base Electrolyte |
|:---:|:---:|:---:|:---:|:---:|:---:|
| | | Ra | Rz | | |
| 1 | 0 | $1.4 \pm 0.4$ | $16.6 \pm 5.0$ | 95 | |
| 2 | 5 | $1.0 \pm 0.2$ | $16.1 \pm 5.0$ | 100 | $KOH + Na_2SiO_3 9H_2O + Na_6P_6O_{18}$ |
| 3 | 10 | $1.2 \pm 0.2$ | $16.5 \pm 3.0$ | 115 | |
| 4 | 15 | $1.1 \pm 0.1$ | $16.5 \pm 3.0$ | 120 | |
| 5 | rubbing-in | $0.6 \pm 0.1$ | $7.8 \pm 2.0$ | 100 | |

### 2.2. Friction and Wear Testing and Analysis

The tribological properties of the PEO coatings were studied in dry conditions on Ball-on-Disc UMT-3 tribometer (CETR (Td Bruker), USA); all tests were performed ac-cording to ASTM G99 [37]. A ball from tungsten carbide with diameter of 10 mm and 72 HRC hardness was pressed by a normal load P = 10 N against the rotating disc with PEO coating (Figure 1b). The sliding velocity of the disc was V = 0.2 m/s, and the wear track (nominal) radius was R = 9 mm. Coatings were tested on friction path S = 1000 m at room temperature T~23 °C and relative humidity 45–50%. Before testing, in order to remove contaminants from the surface of the sample and the ball, they were wiped with ethyl alcohol.

The ball holder was attached to a 2D force sensor which continuously measured normal load and friction force. Note that the normal load was transferred to the force sensor through a mechanical screw transmission (similar to machines for stretching samples), and the lower sample (disk) was driven by a stepper motor that provides the rotation speed

specified by the operator. The coefficient of friction was calculated as the ratio of the friction force to the normal load.

After testing, the worn volume of the entire friction groove was measured using an S Neox optical profilometer (Sensofar, Barcelona, Spain). Specific wear rates were calculated from dividing the wear volume loss by the total sliding distance and load. Three tests were conducted for each coating. In the case of inhomogeneity of the friction track, in-depth direct measurement of worn volume with a profilometer is much more accurate than recommended in ASTM G-99. The profilometer software allows to calculate the entire remote volume below the reference plane (reference level), which is set on a non-worn surface. Note that the wear calculation in ASTM G-99 relies on the assumption of an ideal worn hole geometry, so measurements of the width of the friction track or mass loss are suggested. However, accurate measurement of weight loss is very difficult due to the porosity of the PEO coating material. After measuring the wear, the coatings were examined under the SEM microscope in order to determine their main failure mechanism.

### 2.3. Microscopic Study

The study of the morphology and elemental composition of the surface of the samples was conducted on a scanning electron microscope (SEM) Quanta 650 (FEI, Hillsboro, OR, USA). It was equipped with EDAX analytical equipment, including an energy dispersive X-ray spectral microanalyzer EDS. We used a backscattered electron detector with an accelerating voltage of up to 20 kV. Since the studied ceramic PEO coatings are dielectrics and have low stability under the action of an electron beam, a special low-vacuum microscope mode was chosen during operation. In this mode, rarefied water vapor at a pressure of 15–20 Pa was used as a working medium. The surface of samples with PEO coatings and balls (counter-parts) was studied both in the initial state and after friction tests. In addition, to identify the wear mechanisms, the cross-sections of the coated samples were studied after testing. The samples were cut on an Accutom-5 programmable cutting machine (Struers, Ballerup, Denmark). Grinding and polishing were performed using TegraPol-25 complex with an automatic sample rotator TegraForce-5 (Struers, Ballerup, Denmark).

## 3. Results

### 3.1. Plasma Electrolytic Oxidation (PEO)

The surface of the AlZn5.5MgCu aluminum alloy was treated by PEO method in prepared modified electrolytes to obtain an oxide coating containing aluminum oxides and quasicrystals. Figure 2 shows the corresponding parameters of the electrochemical process in the "potential-time" coordinates for the anode and cathode voltages.

The process of coating formation by the PEO method includes four main stages [17,18,38], as shown in Figure 2. At the beginning of processing, the voltage rapidly increases to a breakdown voltage of about 140 V. During this rather short step (fractions of a second), an amorphous oxide coating is formed by oxidizing the aluminum substrate, as described in [39]; at this stage, only bubbles form and no sparks are observed. In the second stage, which lasts up to 2–5 min, the voltage slowly increases to a breakdown voltage of about 440–500 V, depending on the electrolyte used. At this stage, due to the low breakdown power, small sparks are formed, evenly distributed over the entire surface to be treated. At the third stage, the power and brightness of the sparks on the substrate surface increase, and they gradually transform from sparks into microarc discharges, the power of which is already sufficient for the transformation of the oxide coating from amorphous to crystalline [17]. This stage is the most productive for the formation of coatings, and the longer it lasts, the greater the thickness of the coatings. At the fourth stage, strong arc discharges are created due to an increase in the thickness of the coating and its dielectric properties, which leads to the burning of the coating down to the substrate. As a rule, it is not recommended to bring the coating formation process to the fourth stage, as a result of which our experiments ended with the formation of coatings at the third stage, in order to ensure the best quality of coating formation.

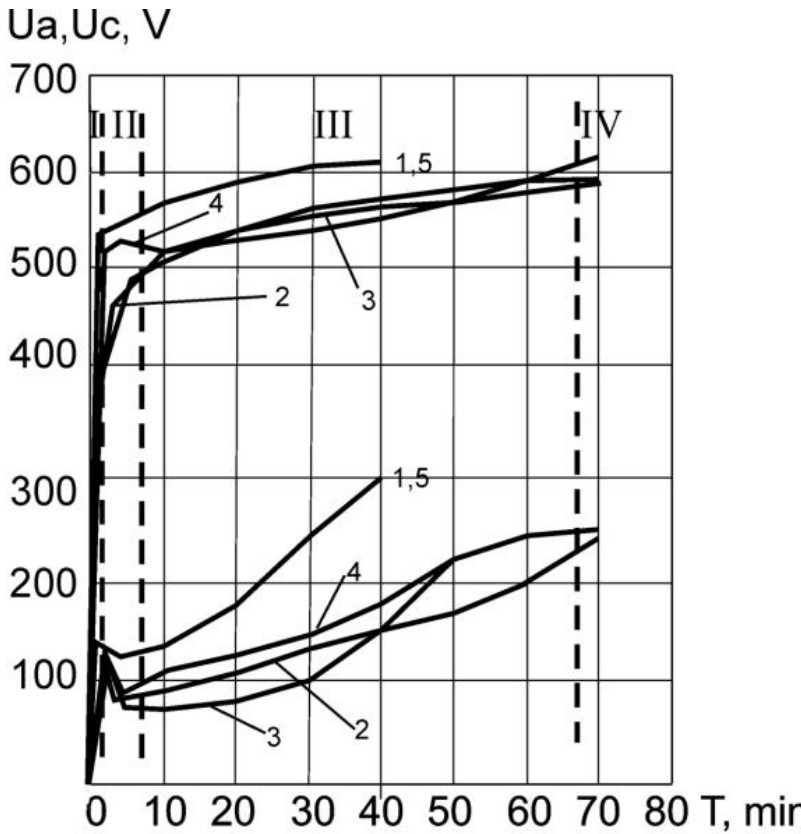

**Figure 2.** "Potential–time" curves for anode and cathode voltages for the PEO process on AlZn5.5MgCu alloy in electrolytes with quasicrystals content (%): 1, 5—0%; 2—0.5%; 3—1.0%; 4—1.5%.

The formation process of PEO coatings in a modified electrolyte with the addition of Al-Cu-Fe quasicrystals is characterized by a slight decrease in the potentials of the anode (Ua) and cathode (Uc) voltages compared to the unmodified electrolyte (the offset of about 30–40 V of the anode voltage and 20–30 V of cathode voltage). The samples formed in the base electrolyte (codes 1 and 5) without quasicrystals were formed at an increased current density (15 A/dm$^2$ instead of 10 A/dm$^2$), as a result of which their formation time was shorter (40 min instead of 70 min) because of higher velocity of the formation and reaching the total coating thicknesses of 150 μm.

It should be noted that no special effect of the concentration of quasicrystals in the electrolyte on the process of coating formation was observed; however, at a higher concentration (1.5%), the color of the formed coating became darker.

### 3.2. Friction and Wear Test

Figure 3 shows typical curves for recording the coefficient of friction as a function of time. It should be noted that for all samples, there is a steady state characterized by a constant coefficient of friction. During the running-in stage, its value increases from 0.2–0.3 to 0.8–1.2 depending on the type of coating modification. Typical 3D pictures of the wear track and the worn surface of the ball are shown in Figure 4.

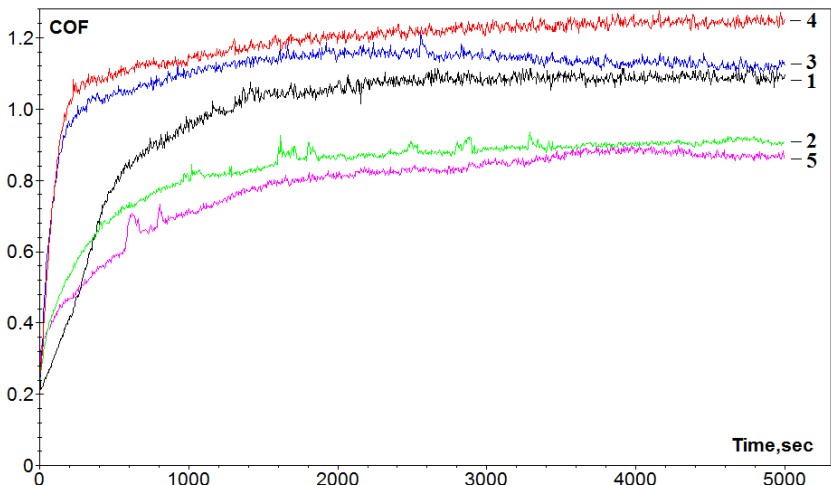

**Figure 3.** The dependence of the coefficient of friction (COF) vs. test time, where 1, 2, 3, 4 and 5—sample codes.

Table 2 presents the results of measuring the steady-state coefficient of friction, volumetric wear and hardness measured at the surface. It is possible to note the correlation between the friction coefficient and wear (with the exception of Sample 4). However, there is no clear relationship between hardness and wear resistance. It should be noted that the hardness values are significantly lower than the average for similar ceramics, which is explained by the structure of the surface layer of the coatings (similar results for PEO coatings were obtained in [40]).

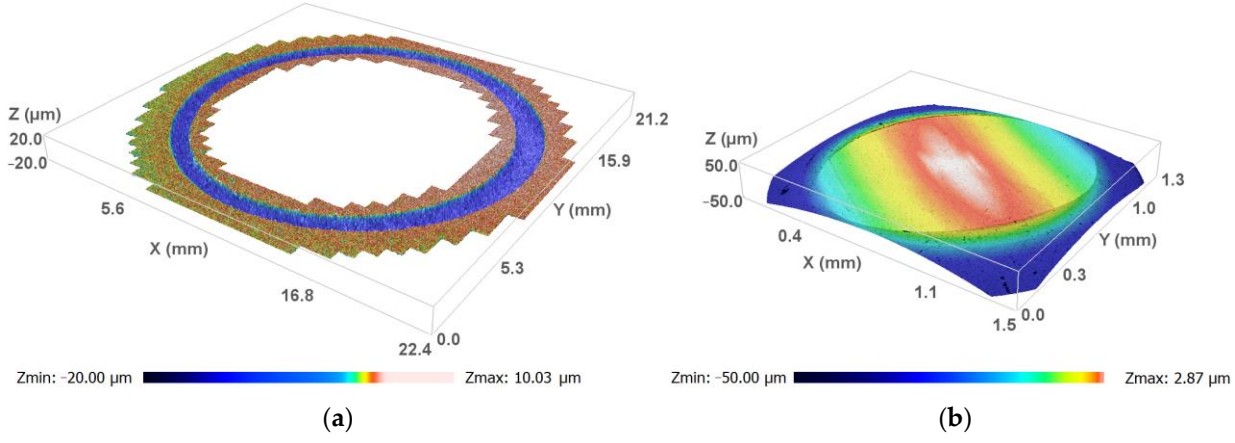

(**a**)                (**b**)

**Figure 4.** The typical 3D profile (Sample 3) of the wear track for disc samples (**a**) and wear scar for ball (**b**).

**Table 2.** The tribological test results.

| Sample Codes | Coefficient of Friction | Disc Volume Losses, mm$^3$ | Disc Specific Wear Rate, mm$^3$/Nm | Ball Volume Losses, mm$^3$ | Hardness, HV |
|---|---|---|---|---|---|
| 1 | $1.10 \pm 0.03$ | $0.434 \pm 0.020$ | $8.7 \pm 0.04 \times 10^{-6}$ | $0.0048 \pm 0.0008$ | $365 \pm 50$ |
| 2 | $1.03 \pm 0.08$ | $0.131 \pm 0.024$ | $2.6 \pm 0.05 \times 10^{-6}$ | $0.0066 \pm 0.0010$ | $300 \pm 30$ |
| 3 | $1.13 \pm 0.04$ | $0.481 \pm 0.089$ | $9.6 \pm 1.80 \times 10^{-6}$ | $0.0134 \pm 0.0012$ | $240 \pm 30$ |
| 4 | $1.17 \pm 0.06$ | $0.310 \pm 0.007$ | $6.2 \pm 0.01 \times 10^{-6}$ | $0.0137 \pm 0.0009$ | $250 \pm 50$ |
| 5 | $0.92 \pm 0.03$ | $0.024 \pm 0.001$ | $0.5 \pm 0.002 \times 10^{-6}$ | $0.0048 \pm 0.0009$ | $350 \pm 50$ |

### 3.3. Microscopic Study

SEM studies of the coated sample surfaces outside the wear track showed the characteristic porosity (Figure 5). The pores differ significantly in their configuration and size, which on average ranges from 2 to 15 μm. It is noted that Samples 1 (without quasi-crystals) and 2 (with a minimum content of quasi-crystals) have the most loosened surface and a greater number of small pores. The surfaces of Samples 3 and 4 are less loose, while Sample 4 (with the maximum content of quasi-crystals) is characterized by the highest surface continuity. The initial surface of Sample 5 differs somewhat from the previous ones: smoother regions are noticeable on it due to the rubbing of quasicrystals into it. The zones used for X-ray EDS analysis (Table 3) are indicated at the photos In Tables 3 and 4, elements related to quasicrystals and the material of the ball are highlighted in gray. It follows from the results that the quasicrystals are concentrated inside the pores.

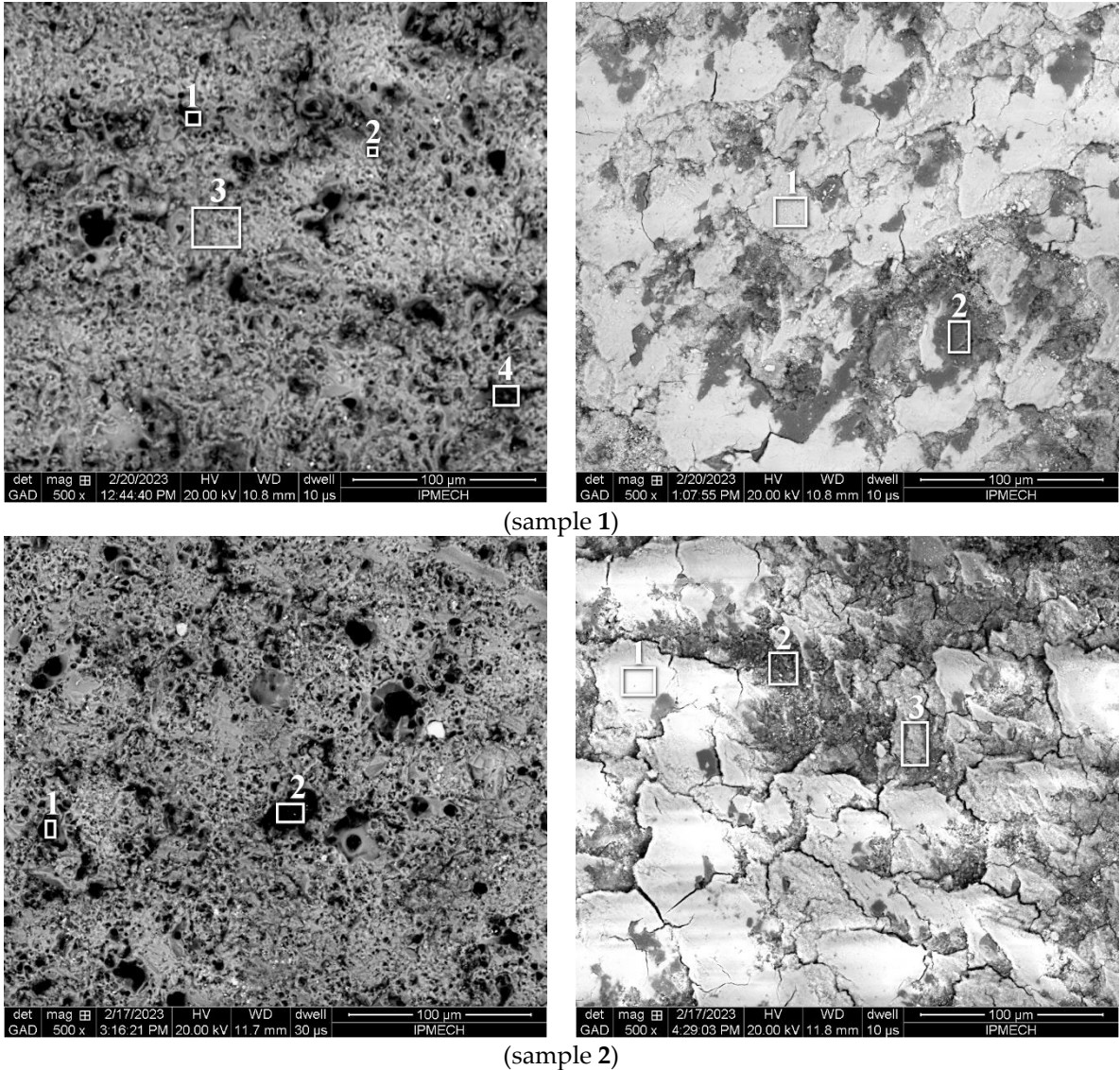

(sample **1**)

(sample **2**)

**Figure 5.** *Cont.*

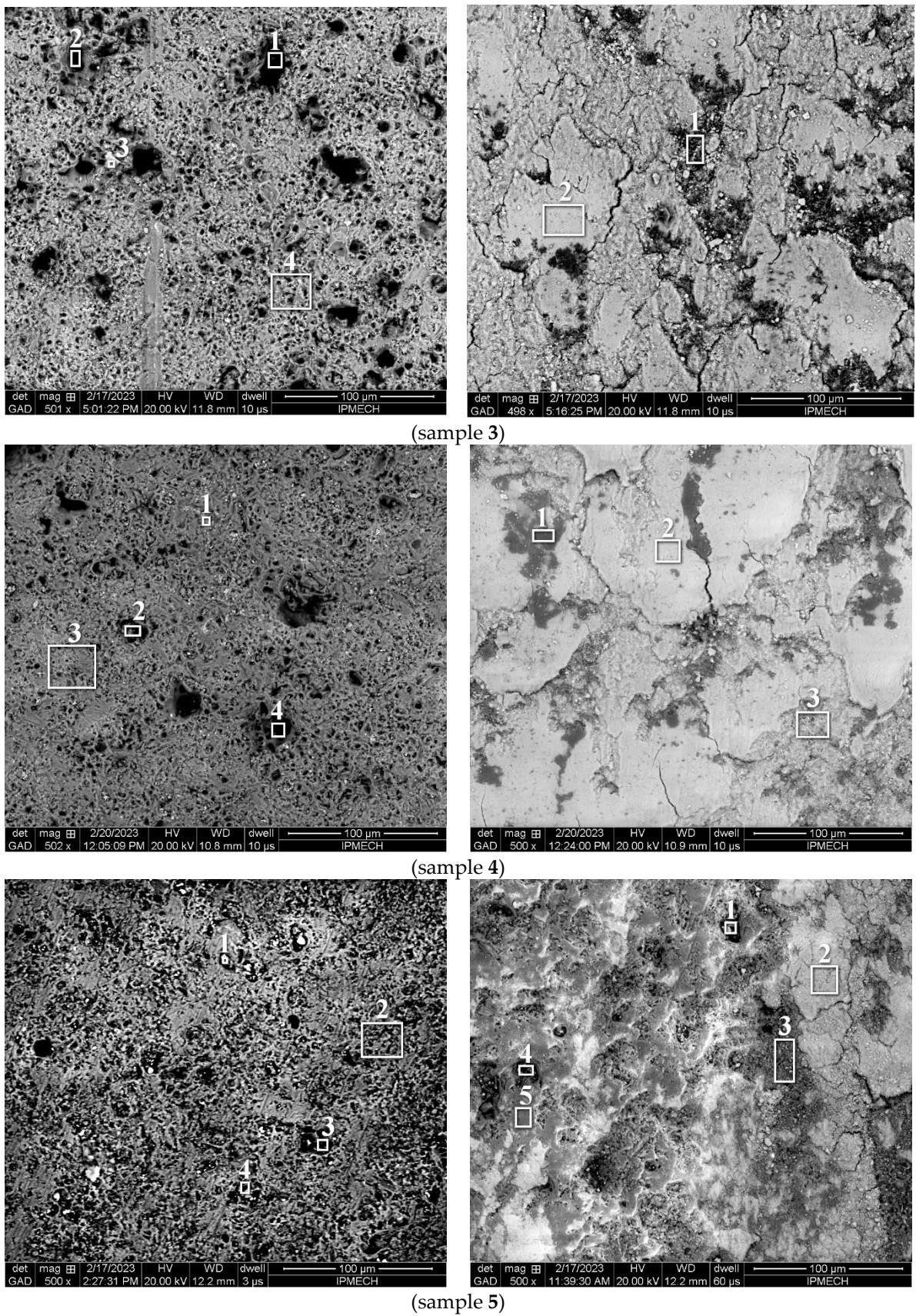

**Figure 5.** Photos of the surface of Samples (**1**–**5**) outside (**left**) and inside (**right**) the friction track; magnification 500.

**Table 3.** Chemical content at the surface outside the friction track (in percent).

| Sample | Zone | C | O | Na | Mg | Al | Si | P | K | Mn | Fe | Cu | W | Zn |
|---|---|---|---|---|---|---|---|---|---|---|---|---|---|---|
| 1 | 1 | 36.41 | 36.40 | 1.51 | 0.66 | 13.18 | 3.87 | 1.97 | 1.48 | 0.36 | - | - | 0.68 | 3.43 |
| | 2 | 12.26 | 39.61 | 1.82 | 0.81 | 35.52 | 2.82 | 2.63 | 1.55 | 0.41 | - | - | 0.38 | 2.19 |
| | 3 | 13.28 | 48.91 | 3.13 | 0.58 | 4.93 | 22.46 | 3.20 | 3.12 | 0.04 | - | - | 0.14 | 0.21 |
| | 4 | 46.91 | 33.06 | 3.84 | 0.48 | 6.89 | 2.35 | 2.41 | 3.03 | 0.13 | - | - | 0.28 | 0.62 |
| | total | 18.16 | 42.99 | 3.72 | 0.71 | 16.51 | 9.96 | 3.86 | 3.14 | 0.12 | - | - | 0.16 | 0.67 |
| 2 | 1 | 19.60 | 17.08 | 0.96 | 0.26 | 22.92 | 2.08 | 0.71 | 1.62 | 3.03 | 12.70 | 5.62 | 5.01 | 8.41 |
| | 2 | 22.33 | 9.70 | 1.70 | 0.29 | 10.90 | 2.42 | 0.46 | 3.76 | 2.66 | 33.09 | 5.00 | 2.78 | 4.91 |
| | total | 16.52 | 44.45 | 2.93 | 0.95 | 20.44 | 7.79 | 3.08 | 2.20 | 0.10 | 0.35 | 0.49 | 0.18 | 0.52 |
| 3 | 1 | 34.86 | 28.43 | 0.13 | 0.00 | 13.75 | 2.17 | 0.04 | 1.68 | 2.61 | 3.84 | 4.21 | 3.71 | 4.57 |
| | 2 | 8.22 | 9.42 | 1.47 | 0.26 | 8.87 | 1.47 | 1.02 | 1.51 | 4.30 | 35.75 | 8.67 | 5.69 | 13.35 |
| | 3 | 6.26 | 35.22 | 2.60 | 0.33 | 21.63 | 9.25 | 3.83 | 3.20 | 0.33 | 1.69 | 2.32 | 11.23 | 2.11 |
| | 4 | 10.76 | 47.77 | 1.77 | 1.05 | 22.05 | 8.58 | 4.51 | 2.02 | 0.08 | 0.53 | 0.42 | 0.07 | 0.39 |
| | total | 11.04 | 43.79 | 2.79 | 0.89 | 21.90 | 10.16 | 3.79 | 2.59 | 0.18 | 0.87 | 0.89 | 0.31 | 0.80 |
| 4 | 1 | 3.89 | 32.23 | 1.37 | 0.53 | 24.00 | 9.65 | 4.26 | 2.11 | 0.05 | 7.90 | 12.68 | 0.92 | 0.41 |
| | 2 | 47.28 | 35.01 | 0.99 | 0.68 | 9.29 | 1.82 | 3.56 | 0.55 | 0.04 | 0.35 | 0.16 | 0.04 | 0.23 |
| | 3 | 6.59 | 46.79 | 0.91 | 1.01 | 29.33 | 10.25 | 1.86 | 1.67 | 0.12 | 0.75 | 0.46 | 0.08 | 0.18 |
| | 4 | 11.53 | 9.56 | 1.22 | 0.52 | 14.75 | 2.43 | 0.68 | 0.78 | 0.85 | 34.20 | 6.38 | 5.53 | 11.57 |
| | total | 9.95 | 42.28 | 3.39 | 0.76 | 21.97 | 10.90 | 3.97 | 2.60 | 0.10 | 1.96 | 1.10 | 0.19 | 0.83 |
| 5 | 1 | 23.54 | 23.68 | 1.40 | 0.57 | 22.34 | 1.02 | 0.41 | 0.33 | 0.27 | 9.21 | 15.72 | 0.92 | 0.59 |
| | 2 | 10.39 | 52.06 | 0.00 | 1.41 | 30.43 | 3.31 | 1.73 | 0.43 | 0.02 | 0.05 | 0.04 | 0.05 | 0.08 |
| | 3 | 28.78 | 45.28 | 0.84 | 1.42 | 16.13 | 2.49 | 1.77 | 1.05 | 0.08 | 0.94 | 0.72 | 0.32 | 0.18 |
| | 4 | 43.47 | 27.89 | 0.04 | 2.59 | 16.08 | 0.69 | 0.54 | 0.25 | 0.06 | 3.94 | 3.88 | 0.34 | 0.23 |
| | total | 13.59 | 47.28 | 1.09 | 1.42 | 27.41 | 5.00 | 1.86 | 1.39 | 0.06 | 0.19 | 0.27 | 0.08 | 0.36 |

**Table 4.** Chemical content at the surface inside the friction track (in percent).

| Sample | Zone | C | O | Na | Mg | Al | Si | P | K | Mn | Fe | Cu | W | Zn |
|---|---|---|---|---|---|---|---|---|---|---|---|---|---|---|
| 1 | 1 | 5.70 | 48.98 | 0.83 | 0.64 | 18.51 | 6.23 | 2.72 | 1.17 | 0.15 | 0.25 | 0.23 | 13.96 | 0.63 |
| | 2 | 7.17 | 53.41 | 0.38 | 0.96 | 17.92 | 5.68 | 2.58 | 1.00 | 0.11 | 0.22 | 0.21 | 9.85 | 0.51 |
| | total | 3.27 | 43.12 | 1.93 | 0.51 | 21.77 | 6.86 | 3.33 | 1.81 | 0.10 | 0.27 | 0.28 | 15.67 | 1.08 |
| 2 | 1 | 6.73 | 57.69 | 1.18 | 1.16 | 19.23 | 6.43 | 2.42 | 1.24 | 0.11 | 0.21 | 0.18 | 3.20 | 0.22 |
| | 2 | 7.30 | 46.97 | 4.76 | 1.05 | 22.14 | 8.62 | 4.06 | 3.06 | 0.10 | 0.19 | 0.33 | 0.96 | 0.46 |
| | 3 | 4.37 | 42.94 | 2.49 | 0.67 | 21.16 | 7.11 | 3.07 | 2.07 | 0.21 | 0.39 | 0.58 | 13.79 | 1.15 |
| | total | 5.91 | 47.99 | 2.00 | 0.85 | 22.48 | 7.76 | 2.87 | 1.66 | 0.08 | 0.31 | 0.40 | 6.99 | 0.70 |
| 3 | 1 | 1.01 | 43.21 | 0.28 | 1.17 | 44.86 | 3.62 | 1.85 | 0.40 | 0.16 | 0.21 | 0.32 | 1.56 | 1.35 |
| | 2 | 4.08 | 47.17 | 1.73 | 0.69 | 20.56 | 6.99 | 3.07 | 1.47 | 0.12 | 0.44 | 0.54 | 12.18 | 0.96 |
| | total | 3.04 | 42.40 | 1.97 | 0.54 | 21.59 | 7.45 | 3.27 | 1.75 | 0.12 | 0.73 | 0.79 | 15.19 | 1.16 |
| 4 | 1 | 2.14 | 47.91 | 2.15 | 1.66 | 32.29 | 5.35 | 4.85 | 1.15 | 0.12 | 0.25 | 0.28 | 0.86 | 0.99 |
| | 2 | 4.86 | 56.01 | 0.65 | 1.10 | 18.53 | 7.76 | 2.92 | 1.19 | 0.06 | 0.44 | 0.34 | 5.80 | 0.34 |
| | 3 | 2.17 | 38.55 | 2.66 | 0.64 | 23.07 | 7.50 | 3.76 | 2.29 | 0.17 | 0.81 | 1.05 | 15.70 | 1.63 |
| | total | 2.57 | 44.16 | 2.24 | 0.60 | 21.37 | 8.27 | 3.30 | 1.72 | 0.13 | 0.76 | 0.83 | 13.03 | 1.02 |
| 5 | 1 | 26.22 | 37.98 | 0.00 | 0.84 | 24.34 | 1.94 | 1.03 | 0.23 | 0.05 | 3.18 | 3.70 | 0.26 | 0.23 |
| | 2 | 6.17 | 41.04 | 1.11 | 0.86 | 28.06 | 4.96 | 1.58 | 1.44 | 0.13 | 0.34 | 0.61 | 12.35 | 1.35 |
| | 3 | 9.91 | 43.75 | 0.59 | 2.06 | 33.94 | 3.42 | 1.61 | 1.16 | 0.24 | 0.31 | 0.56 | 1.49 | 0.96 |
| | 4 | 45.91 | 21.01 | 0.00 | 0.37 | 17.54 | 0.90 | 0.29 | 0.31 | 0.17 | 5.70 | 6.56 | 0.92 | 0.32 |
| | 5 | 5.12 | 47.08 | 0.59 | 1.15 | 36.92 | 4.98 | 1.95 | 1.14 | 0.06 | 0.08 | 0.08 | 0.31 | 0.54 |
| | total | 5.01 | 46.66 | 0.72 | 0.98 | 32.06 | 5.03 | 1.67 | 1.01 | 0.11 | 0.20 | 0.29 | 5.29 | 0.97 |

After friction tests, significant changes occur. The surface morphology becomes smoother, since during friction a tribofilm is formed in the contact zone, which hides the pores. This film has cracks over the entire surface of the friction track. The chemical analysis (Table 4) showed that it contains chemical elements corresponding to tungsten carbide, which is transferred from the counter-part during friction. In order to determine the presence of quasicrystals on the friction surface, mapping was performed for copper and iron, as well as for tungsten. The results for Sample 5 showing the presence of quasicrystals are presented in Figure 6. Quasi-crystals were also found in a smaller amount on the friction surface of Sample 2. In other cases, the elements of quasi-crystals were not detected.

The surface of the balls after testing was also analyzed. The size of the worn spot fully correlates with the amount of volumetric wear. Particles containing aluminum were fixed at the edge of the wear zone. It can be argued that quasicrystals are not transferred to the surface of the counter-part.

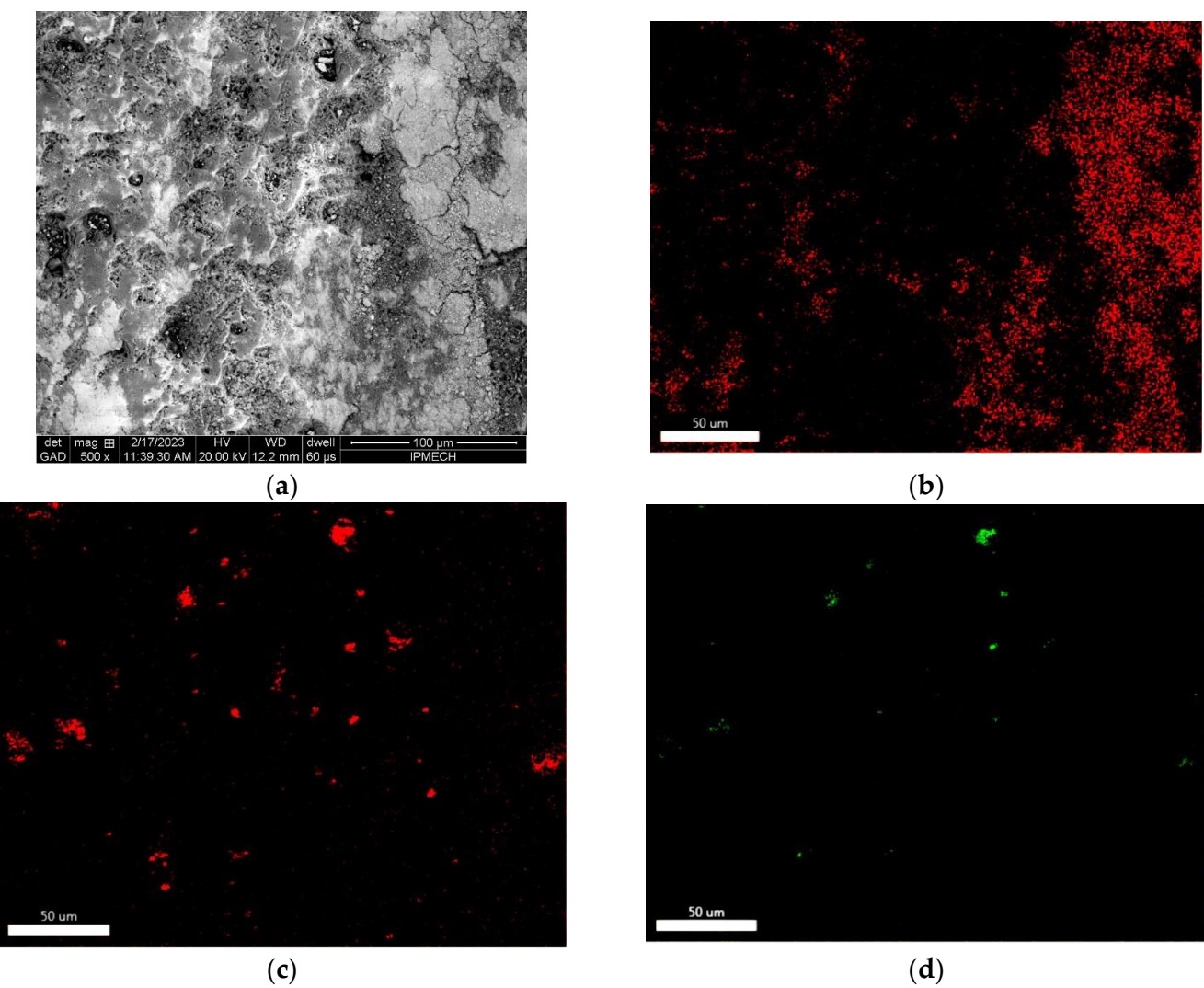

**Figure 6.** Mapping of W (**b**), Fe (**c**), Cu (**d**) at the friction surface (**a**) of Sample 5; magnification 500.

## 4. Discussion

The influence of the modification of ceramic coatings by quasicrystals on the processes of friction and wear is ambiguous, which is associated with two factors: a change in the structure of ceramics when quasicrystals are added to the electrolyte, and also with the presence of friction-reducing quasicrystals on the contact surface. To eliminate the first factor, the technology of rubbing quasi-crystals into the surface during final treatment was used. With this technology, only surface pores of different sizes are filled with quasicrystals, but their influence on the composition and structure of the coating is excluded. If we compare the amount of the modifier in the pores and, in general, on the surface, with the value of the steady friction coefficient, then we can conclude that a small concentration makes it possible to reduce the friction coefficient compared to the base ceramic coating. This is demonstrated in Samples 2 and, especially, 5. In the case of Sample 5, the decrease in the friction coefficient is also associated with a decrease in roughness due to additional surface treatment using quasicrystals (Ra of Sample 1 was 1.4 μm, and for Sample 5 it was 0.6 μm). This is a promising result, since it allows minimizing the consumption of a rather expensive modifier. With an increase in the concentration of quasicrystals in the electrolyte, coatings demonstrate an increase in the friction coefficient.

At the beginning of the running-in process, all coatings show relatively low friction. With the appearance of wear particles, the coefficient of friction increases. The steady state is usually characterized by the steady volume of the third body. In our case, as the results of chemical analysis show, the tribofilm consists of ball wear products. The greater the coefficient of friction, the more tungsten carbide on the surface (see Table 4) and in the form of a non-continuous film. The lower the coefficient of friction, the more areas without a film are on the surface of the coating. Chemical elements corresponding to quasicrystals were detected in these areas.

Considering that the contact of the ball with the coating occurs through the tribofilm, it is important to identify the wear mechanism of the coatings. The results of the study of cross-sections show that there are microcracks associated with pores in the coating (see Figure 7). The development of microcracks due to stress concentration leads to delamination and removal of an element of the near-surface layer. From the photos inside and outside the friction track, we can see that the surface of the coating on the friction track becomes less even. The effect of gloss on the friction track, observed visually, is associated with the presence of the tribofilm. Figure 7 shows photographs of Sections of Coatings 4 and 5, which differ significantly in the value of the friction coefficient and volumetric wear. In the case of Coating 5, there are no significant differences in the structure of the coating under the friction track and away from it. In Coating 4, there is a tendency to merge pores and form cracks coming from the surface. With such a wear mechanism, the magnitude of the friction force is important, since it creates tensile stresses on the surface that are favorable for the development of cracks. With the exception of Sample 4, wear correlates with the coefficient of friction. Coating Sample 4 is more wear resistant than Samples 1 and 3; its coefficient of friction is higher, and wear is less. This is probably due to the influence of the maximum amount of the modifier in the electrolyte on the structure of the material, which leads to the presence of quasi-crystals inside the coating (Figure 7e,f). The decrease in the number of quasicrystals closer to the substrate is associated with a decrease in the number of pores in the PEO coating.

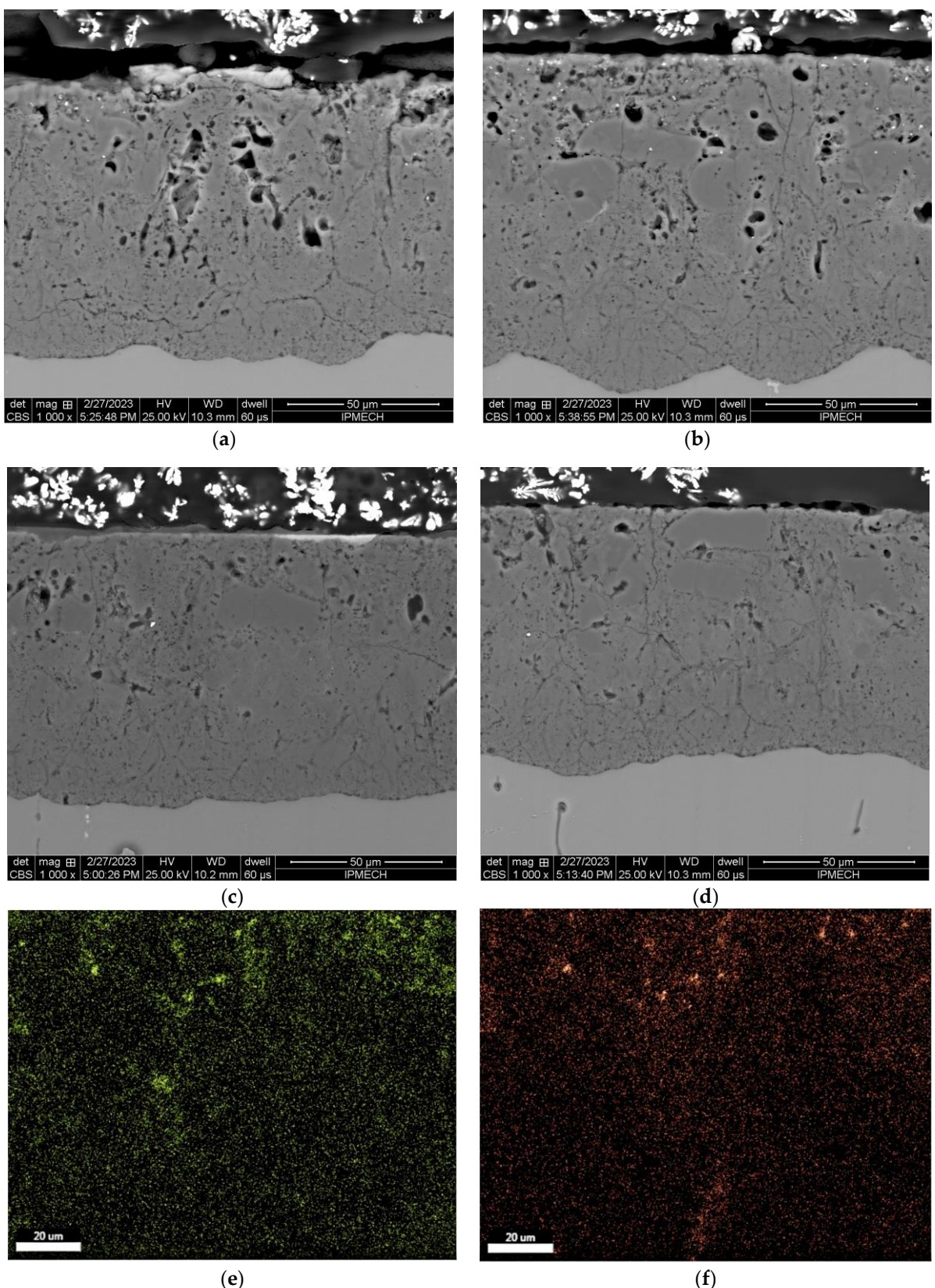

**Figure 7.** Cross-sections. The images of samples under wear tracks (**a**,**c**) and under initial surfaces (**b**,**d**); Sample codes 4 (**a**,**b**) and 5 (**c**,**d**). Mapping of Cu (**e**) and Fe (**f**) for Sample 4 (under initial surface).

## 5. Conclusions

Because of using the technology with the addition of quasicrystals to the electrolyte during the formation of PEO coatings, ceramic coatings containing quasicrystals in the pores were obtained. The structure of the coatings depends on the content of the modifier in the electrolyte. The technology of rubbing quasicrystals into the surface of pure ceramic coatings also leads to the concentration of quasicrystals in pores without changing the structure of the material.

It was determined that the smallest amount of quasi-crystals is optimal for reducing friction and wear (compared to coatings with zero and relatively high modifier content in the electrolyte, 1 and 1.5 percent). The processes of friction and wear are significantly affected by the tribofilm, which consists of wear products of the ball, the material of which is tungsten carbide. The mechanism of wear of coatings is brittle fracture at the microlevel with subsequent separation of particles from the surface.

**Author Contributions:** Conceptualization, E.V.T., A.V.M. and V.N.M.; investigation, E.V.T., A.V.M. and O.O.S.; writing—original draft preparation, E.V.T.; writing—review and editing, E.V.T., V.N.M. and A.V.M.; visualization, E.V.T., O.O.S. and A.V.M.; project administration, E.V.T.; material processing V.N.M. All authors have read and agreed to the published version of the manuscript.

**Funding:** The present study was supported by the Ministry of Science and Higher Education within the framework of the Russian State Assignment under contract No. 123021700050-1 (FFGN-2023-0005).

**Institutional Review Board Statement:** Not applicable.

**Informed Consent Statement:** Not applicable.

**Data Availability Statement:** Not applicable.

**Acknowledgments:** The authors are grateful to O. Neyaglov for providing Al-Cu-Fe quasicrystals.

**Conflicts of Interest:** The authors declare no conflict of interest.

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
