# Peer review of "Processing and Tribological Properties of PEO Coatings on AlZn5.5MgCu Aluminium Alloy with Incorporated Al-Cu-Fe Quasicrystals"

_ceramics, doi:10.3390/ceramics6020049_

Round 1

Reviewer 1 Report

Dear authors,

Your work is quite interesting, but improvements are needed.

1. It is necessary to add the numerical results obtained in work to the abstract.

2. The research topic described in the introduction is very superficial. It is necessary to show the novelty of the work and clarify by what methods the analysis of the results was carried out.

3. P. 2. Firms and countries of manufacture must be added to the equipment and chemicals used.

4. In the chapter ‘‘Results,’’ it is necessary to present graphs of the PEO process's current-time and voltage-time. It is necessary to describe the stages of coating formation, as well as the effect of the concentration of quasicrystals on the features of coating formation. As a new and interesting work, I propose to include the following work in the list of cited literature https://doi.org/10.1016/j.corsci.2022.110604.

5. It is not very clear why the quasicrystals located in the surface layer of the coating have such a significant effect on the coefficients of friction and the microhardness of the coating. Or did the introduction of quasicrystals occur throughout the entire depth of the coating? If so, why are they not visible in the cross-sections?

6. In addition, you say that the temperatures on the surface of the workpiece during the PEO process reach up to 12000 K. At this temperature, the quasicrystals should melt and be part of the base coating in the form of oxides of aluminum, copper, and iron.

7. The work is written in a very complex form for readers to understand. It isn't easy to trace the logical chain. I think the work should be reworked, and the results and discussion combined.

Author Response

Thank you very much for your appreciation of our research and helpful comments. Changes have been made to the text highlighted in red. Below are the comments and our responses.

  1. It is necessary to add the numerical results obtained in work to the abstract.

The results are added in the new version

  1. The research topic described in the introduction is very superficial. It is necessary to show the novelty of the work and clarify by what methods the analysis of the results was carried out.

The last paragraph of the Introduction has been changed.

  1. P. 2. Firms and countries of manufacture must be added to the equipment and chemicals used.

It is done.

  1. In the chapter ‘‘Results,’’ it is necessary to present graphs of the PEO process's current-time and voltage-time. It is necessary to describe the stages of coating formation, as well as the effect of the concentration of quasicrystals on the features of coating formation. As a new and interesting work, I propose to include the following work in the list of cited literature https://doi.org/10.1016/j.corsci.2022.110604.

New Figure 2 is included to the paper, as well as several references (marked by red font)

  1. It is not very clear why the quasicrystals located in the surface layer of the coating have such a significant effect on the coefficients of friction and the microhardness of the coating. Or did the introduction of quasicrystals occur throughout the entire depth of the coating? If so, why are they not visible in the cross-sections?

Mapping for the quasicrystals elements in cross-section of sample 4 are in Figure 7 in the revised version.

  1. In addition, you say that the temperatures on the surface of the workpiece during the PEO process reach up to 12000 K. At this temperature, the quasicrystals should melt and be part of the base coating in the form of oxides of aluminum, copper, and iron.

Lines 86-95 contain a discussion of this issue. Thank you for bringing this important point to our attention.

  1. The work is written in a very complex form for readers to understand. It isn't easy to trace the logical chain. I think the work should be reworked, and the results and discussion combined.

Usually we combine the Results and the Discussion into a single section, but in this case it is inconvenient, since in the Discussion we appeal simultaneously to the entire set of results. We've added some text to the Results.

Reviewer 2 Report

This work study the influence of Al-Cu-Fe quasicrystal on the wear and friction performance of PEO coating on AlZn5.5MgCu aluminum alloy. The results shows that the existence of quasicrystal could dramatic influence the friction. However, there are some problems need to be addressed and several improvements can be made. Here are my comments:

1.The language needs to be improved. Many sentences have grammar error or do not conform to the standards of academic writing. For example: lines 24-26, line 45-47, line 66, line 77 (should not use bad to describe roughness), line 98 (45-50%), line 159, line 194-195 (It is not proper to describe the result as a good result), line 203-204, line 205 (free areas).

2.Quasicrystal particles used in this paper is not described or characterized. What is the specific chemical composition of Al-Cu-Fe system quasicrystals? What is the shape and size of the particles?

3.It is mentioned that quasicrystals were rubbed into the surface. What is the specific method used to charry out this process? Please add this in the method section.

4.It is mentioned that direct measurement of worn volume with a profilometer is much more accurate. Please provide the data analyze method.

5.It is mentioned that ceramic PEO coating have low stability under electron beam. It might have charging effect under SEM but it is hard to believe that metal oxide is unstable under electron beam irradiation. Moreover, what do you mean by surface metallization?  

6.Figure 2 shows the coefficient of friction as a function of time. How do you measure the coefficient of friction? Detailed method should be described.

7.Figure 3 (a) and (b) lack color scale bar.

8.It is mentioned that the hardness is much lower than normal. This is because the surface is porous. The number measured is not the real hardness. I don’t think it is proper to report these results.

9.Line 156 X-ray spectral analysis should be EDS analysis.

10.Table 3 and Table 4 contains too much information. While they are not being explained enough. If you use EDS to get these information, I suggest to use the EDS mapping images instead and combine them with figure 4. You can put these tables in the supporting information.

11.Method used to make the cross-section sample should be described in method section.

12.It is mentioned that existence of quasicrystals in the electrolyte would change the ceramics structure. How do you get this conclusion? Please provide more evidence or explanation. 

Author Response

Thank you very much for your appreciation of our research and helpful comments. Changes have been made to the text highlighted in red. Below are the comments and our responses.

1.The language needs to be improved. Many sentences have grammar error or do not conform to the standards of academic writing. For example: lines 24-26, line 45-47, line 66, line 77 (should not use bad to describe roughness), line 98 (45-50%), line 159, line 194-195 (It is not proper to describe the result as a good result), line 203-204, line 205 (free areas).

We have changed the phrases marked in the comment.

2.Quasicrystal particles used in this paper is not described or characterized. What is the specific chemical composition of Al-Cu-Fe system quasicrystals? What is the shape and size of the particles?

Additional information about quasicrystals is given in the first paragraph of Section 2, and in Figure 1a

3.It is mentioned that quasicrystals were rubbed into the surface. What is the specific method used to charry out this process? Please add this in the method section.

The text was added (lines 100-103).

4.It is mentioned that direct measurement of worn volume with a profilometer is much more accurate. Please provide the data analyze method.

The last paragraph of section 2.2 is written in more detail (in terms of the discussion of wear measurement).

5.It is mentioned that ceramic PEO coating have low stability under electron beam. It might have charging effect under SEM but it is hard to believe that metal oxide is unstable under electron beam irradiation. Moreover, what do you mean by surface metallization? 

We removed the phrase about metallization. It is really inappropriate.

6.Figure 2 shows the coefficient of friction as a function of time. How do you measure the coefficient of friction? Detailed method should be described.

The detailed method is described (lines 128-134)

7.Figure 3 (a) and (b) lack color scale bar.

It is done

8.It is mentioned that the hardness is much lower than normal. This is because the surface is porous. The number measured is not the real hardness. I don’t think it is proper to report these results.

These results are of value for comparative analysis. Other researchers have also obtained similar values when analyzing cross-sections of ceramic coatings.

9.Line 156 X-ray spectral analysis should be EDS analysis.

It is done

10.Table 3 and Table 4 contains too much information. While they are not being explained enough. If you use EDS to get these information, I suggest to use the EDS mapping images instead and combine them with figure 4. You can put these tables in the supporting information.

We would not like to remove these tables from the text, since quantitative values are important. For better visualization, we highlighted the columns of elements corresponding to quasi-crystals and counter-body.

11.Method used to make the cross-section sample should be described in method section.

Lines 158-160 contain this description.

12.It is mentioned that existence of quasicrystals in the electrolyte would change the ceramics structure. How do you get this conclusion? Please provide more evidence or explanation.

Some information is in new Section 3.1. Also we added mapping for the quasicrystals elements in cross-section of sample 4 (Figure 7).

Round 2

Reviewer 1 Report

Good job!!!

Reviewer 2 Report

The authors did a good job in solving most of the comments. I have three more suggestions.

1: Please carefully check the grammar and language of the revised paragraphs. 

2: I suggest adding element names on the EDS mapping images. 

 3: I am confused of new figure 7 e and f. Where is the region being tested? It is described as Mapping of Cu (e) and Fe (f) for sample 4 9under initial surface). (By the way, this sentence itself has three errors. There should be a ‘and’ between 4 and 9. There should be a space between 9 and under. You should delete the ‘)’ after surface. There are also many similar errors in the current manuscript) What do you mean by under initial surface? Is the EDS mapping carried out on the cross section or on the top surface?